# Impact of the Universal Implementation of Adolescent Hepatitis B Vaccination in Spain

**DOI:** 10.3390/vaccines12050488

**Published:** 2024-05-01

**Authors:** Angela Domínguez, Ana Avellón, Victoria Hernando, Núria Soldevila, Eva Borràs, Ana Martínez, Conchita Izquierdo, Núria Torner, Carles Pericas, Cristina Rius, Pere Godoy

**Affiliations:** 1Department of Medicine, Universidad de Barcelona, 08036 Barcelona, Spain; angela.dominguez@ub.edu (A.D.); eva.borras@gencat.cat (E.B.); ntg@ub.edu (N.T.); cpericas@aspb.cat (C.P.); 2CIBER Epidemiología y Salud Pública (CIBERESP), Instituto de Salud Carlos III, 28029 Madrid, Spain; aavellon@isciii.es (A.A.); a.martinez@gencat.cat (A.M.); crius@aspb.cat (C.R.); pere.godoy@gencat.cat (P.G.); 3Hepatitis Unit, National Centre of Microbiology, Instituto de Salud Carlos III, 28222 Madrid, Spain; 4Centro Nacional de Epidemiología, Instituto de Salud Carlos III, 28029 Madrid, Spain; vhernando@isciii.es; 5CIBER Enfermedades Infecciosas (CIBERINFEC), Instituto de Salud Carlos III, 28029 Madrid, Spain; 6Agència de Salut Pública de Catalunya, 08005 Barcelona, Spain; conchita.izquierdo@gencat.cat; 7Agència de Salut Pública de Barcelona, 08023 Barcelona, Spain; 8Institut de Recerca de l‘Hospital de la Santa Creu i Sant Pau (IRB Sant Pau), 08041 Barcelona, Spain; 9Institut de Recerca Biomédica de Lleida (IRBLleida), 25006 Lleida, Spain

**Keywords:** hepatitis B, vaccination strategy, impact evaluation

## Abstract

The aim of this study was to analyse the impact of the introduction of universal adolescent HBV vaccination on the incidence of acute hepatitis B virus (HBV) infections. Acute HBV cases reported to the Spanish National Epidemiological Surveillance Network between 2005 and 2021 were included. For regions starting adolescent vaccination in 1991–1993 and in 1994–1996, HBV incidence rates were compared by calculating the incidence rate ratio (IRR) and 95% confidence interval (CI). We also analysed the 2017 Spanish national seroprevalence survey data. The overall acute HBV incidence per 100,000 persons was 1.54 in 2005 and 0.64 in 2021 (*p* < 0.001). The incidence in 2014–2021 was lower for regions that started adolescent vaccination in 1991–1993 rather than in 1994–1996 (IRR 0.76; 95% CI 0.72–0.83; *p* < 0.001). In the 20–29 age group, incidence in regions that started adolescent vaccination in 1991–1993 was also lower (IRR 0.87; 95% CI 0.77–0.98; *p* = 0.02 in 2005–2013 and IRR 0.71; 95% CI 0.56–0·90; *p* < 0.001 in 2014–2021). Anti-HBc prevalence in the 35–39 age group was lower in the regions that started vaccination earlier, although the difference was not statistically significant (*p* = 0.09). Acute HBV incidence decreased more in the young adult population in regions that began adolescent vaccination earlier. Maintaining high universal vaccination coverage in the first year of life and in at-risk groups is necessary to achieve HBV elimination by 2030.

## 1. Introduction

Hepatitis B virus (HBV) is responsible for most of the chronic hepatitis disease burden caused by viral hepatitis variants worldwide, with vaccination as the cornerstone of global elimination initiatives. HBV displays 10 different genotypes (A to J) with a DNA variation of 8% between genotypes and subgenotypes which differ by 4%. Genotype distributions vary geographically. HBV genotypes are associated with variable disease courses, liver disease severity, and treatment outcomes [1]. WHO member states have endorsed goals for HBV elimination as a public threat by 2030 [2]. The administration of three doses of HBV vaccines to infants in the first year of life is a challenge, but also an opportunity to eliminate HBV, with a first dose recommended as soon as possible after birth (within 24 h) [3]. This is currently considered the best strategy to eliminate HBV as a public health threat by 2030 [2].

HBV is spread in several ways: through sexual contact; by sharing needles, syringes, or other drug-injection equipment; and perinatally from mother-to-child at birth [1]. The typical mode of HBV transmission varies according to the prevalence of infection. In high- and intermediate-endemicity areas, the predominant route of transmission is perinatal or horizontal during childhood, whereas in low-prevalence regions, such as Spain, transmission occurs mainly via injected drug use and high-risk sexual behaviours [4].

Sexual transmission during heterosexual and male-to-male sexual contact is a major cause of HBV infection in adults in low-HBV-endemicity regions [5]. The injection of illicit drugs is a major nonmedical source of percutaneous exposure to HBV, and globally incidence rates of 10–21% per year have been reported among people who inject intravenous drugs [6]. In addition, selling sex for money or drugs also contributes to transmission among drug users [7].

With the widespread vaccination of healthcare workers and the implementation of standard precautions in healthcare setting, the incidence of new infections has declined in this collective. Workers whose job duties (policemen, firefighters and prison workers) present opportunities for exposure to blood may be at increased risk for HBV infection, but in practice the prevalence of HBV infection in those workers does not differ from that of the general population [8].

The first HBV vaccines, derived from human plasma, became commercially available in 1982, but were replaced by a second generation of genetically engineered HBV vaccines developed in 1986 with recombinant HBV surface antigen (HBsAg) [9]. The development of recombinant DNA technology to express HBsAg in yeast, and later also in mammalian cells, offered the potential to produce large quantities of vaccine. Recombinant hepatitis B vaccines are highly immunogenic [10], although older age, persons with risk factors such as smoking, genetic factors, or underlying chronic conditions may result in lower response rates [11]. Genotypes and certain subgenotypes have distinct geographical distributions and are important in both the clinical manifestation of infection and the response to antiviral therapy [12]. Some A genotypes have been found to be significant independent protectors against chronic infection and more responsive to treatment [13,14]. However, no statistically significant differences in HBV genotype distributions according to hepatitis B vaccination history have been reported [15].

In September 2015, the United Nations General Assembly adopted the 2030 Agenda for Sustainable Development, which enshrines 17 goals and associated targets. Of particular relevance is target 3.3: “By 2030, end the epidemics of AIDS, tuberculosis, malaria and neglected tropical diseases and combat hepatitis, water-borne diseases, and other communicable diseases” [16]. Administration of three doses of hepatitis B vaccines to infants in the first year of life is currently considered the best strategy to achieve the elimination of hepatitis B as a public health threat in 2030 [2,17]. 

A plasma-derived HBV vaccination programme was launched in Spain in 1984 focused on at-risk groups and newborns of HBsAg-positive mothers. However, the impact on disease incidence and long-term outcomes was limited. The availability of effective, safe, and less costly recombinant vaccines thus opened the way for mass vaccination programs to protect whole population.

In 1996, HBV vaccination for all preadolescents and younger adolescents (aged 10–14 years) was introduced in Spain with the first Recommended Immunization Schedule, published by the Interterritorial Council of the National Health System (CISNS). However, HBV vaccination of adolescents (in addition to vaccination of at-risk groups and newborns of HBsAg positive mothers) was rolled out at different times in Spanish regions, starting with Catalonia in 1991 [18], and covering all of the remaining Spanish autonomous regions by 1996. By 2002, the universal vaccination of infants in the first year of life was implemented in all Spanish regions [19], and by 2004, it was included in the Recommended Immunization Schedule of the CISNS. The vaccination of adolescents aged 10–14 years was maintained until publication of the CISNS Recommended Immunization Schedule of 2014, prescribing immunization only of infants in their first year of life. Hepatitis B vaccination coverages were high, fluctuating between 75% and 90% in preadolescents and adolescents [20,21] and higher than 95% in infants in their first year of life [22].

Since exploring the results obtained for different vaccination strategies is of potential interest from a public health perspective, the aim of this study was to analyse the impact of the introduction of universal vaccination on acute HBV incidence over a 17-year period, overall, for specific age groups, and for the regions that implemented the universal vaccination of adolescents in the years 1991–1993 versus those that did so later (1994–1996).

## 2. Materials and Methods

A retrospective study was carried out from 2005 to 2021 in Spain. All acute cases of HBV reported to the National Epidemiological Surveillance Network were included.

The clinical case definition of acute HBV was the gradual onset of symptoms such as fatigue, abdominal pain, loss of appetite, intermittent nausea and vomiting, plus at least one of fever, jaundice, and elevated serum aminotransferase levels. A confirmed case was defined as a case that met the clinical case definition confirmed by laboratory testing [23].

Acute HBV case data were collected by age group, sex, and region. Regional population data were obtained from the National Institute of Statistics. Incidence per 100,000 person-years was calculated overall, by sex, and for specific age groups. Incidence was calculated for the 20–29 and 30–39 age groups, because these ranges encompassed the ages in 2005–2021 of persons who were 12 years old in 1991–1996 (12 years was the average age at which vaccination began) (Appendix A); incidence was also calculated for the 40–49 age group as a cohort that was not vaccinated during adolescence.

HBV vaccination of adolescents in Spain was rolled out between 1991 and 1996 in two main phases: in 1991–1993 in Catalonia, the Balearic Islands, Valencia, the Basque Country and La Rioja and in 1994–1996 in Andalusia, Galicia, Murcia, Asturias, Aragon, Castilla-La Mancha, Extremadura, Navarra, the Canary Islands, Cantabria, Castilla Leon and Madrid.

Incidences in regions that started vaccination in 1991–1993 and in those that started adolescent vaccination in 1994–1996 were compared by calculating the incidence rate ratio (IRR) and 95% confidence interval (CI) values; analyses were also performed for two distinct periods: 2005–2013, as a period closer to the start of vaccination, and 2014–2021, as a later period.

Incidence trends were calculated for the 17-year study period, 2005–2021, as a whole, for regions that started vaccination in 1991–1993 and for regions that started vaccination in 1994–1996 using the chi-square test for trend.

The total antibody to HBV core antigen (anti-HBc) results obtained in the 2017 Spanish national seroprevalence survey of representative samples of all the Spanish regions were analysed, separately considering the results of regions that started adolescent vaccination in 1991–1993 and those of the regions that started adolescent vaccination in 1994–1996 [24]. The HBV prevalence rates were compared using the chi-square test.

Analyses were performed using R version 4.3.0 statistical software and OpenEpi version 3.01.

## 3. Results

The acute HBV incidence overall in 2005 per 100,000 persons was 1.54, falling to 0.64 by 2021. The declining trend was significant (*p* < 0.001) overall, in the regions that started adolescent vaccination in 1991–1993 and in the regions that started adolescent vaccination in 1994–1996 (Figure 1).

According to the starting year of vaccination, in 2014–2021 and for all age groups, acute HBV incidence was lower in regions that started adolescent vaccination in 1991–1993 than in regions that started adolescent vaccination in 1994–1996 (IRR 0.76; 95% CI 0.72–0.83), while differences were not statistically significant in 2005–2013. This lower incidence was observed both in men and women in 2014–2021, but only in women in 2005–2013; for men in 2005–2013, incidence was higher in regions than started adolescent vaccination in 1991–1993 (Table 1).

Regarding the 20–29 age group, incidence in regions that started adolescent vaccination in 1991–1993 was lower than in regions that started adolescent vaccination in 1994–1996, both in 2005–2013 (IRR 0.87; 95% CI 0.77–0.98) and in 2014–2021 (IRR 0.71; 95% CI 0.56–0.90). This lower incidence in regions that started adolescent vaccination in 1991–1993 was observed in men in 2014–2021 (IRR 0.68; 95% CI 0.51–0.90) and in women in 2005–2013 (IRR 0.79; 95% CI 0.63–0.98).

In the 30–39 age group, in 2014–21, the incidence in regions that started adolescent vaccination in 1991–1993 was lower than in regions that started adolescent vaccination in 1994–1996, overall (IRR 0.74; 95% CI 0.63–0.86) and both in men and women (IRR 0.74; 95% CI 0.62–0.88 and IRR 0.73; 95% CI 0.54–0.99, respectively). In contrast, in 2005–2013, the incidence in men was higher in regions that started adolescent vaccination in 1991–1993 (IRR 1.14; 95% CI 1.03–1.26). Bear in mind that most people aged 30–39 years in 2005–2013 were not vaccinated in 1991–1996 because they were no longer adolescents at that time.

In the 40–49 age group, no statistically significant differences were observed, neither overall nor by sex or by period. Virtually all persons aged 40–49 years in 2005–2021 had not been vaccinated in 1991–1996 because they were no longer adolescents when the vaccination program was rolled out.

Data from the 2017 Spanish national seroprevalence survey show that anti-HBc prevalence in the 35–39 age group was lower in the regions that launched universal adolescent immunization earlier rather than later, although this difference was not statistically significant (0% vs. 3.6%; *p* = 0.09), probably because of the small size of that age group (Figure 2). In the 25–29 and 30–34 age groups, anti-HBc prevalence was higher in the regions that launched universal adolescent immunization in 1991–1993, although the differences were not statistically significant (2.6% versus 0%; *p* = 0.10 and 3.6% versus 1.2%; *p* = 0.29, respectively). However, in 2017, all persons aged under 34 years had been vaccinated in all regions (i.e., in both those that started vaccination in 1991–1993 and in those that started in 1994–1996).

## 4. Discussion

This study confirms the decrease in acute HBV incidence in all Spanish regions in the period 2005–2021, corroborating previous reports by studies carried out between 1997 and 2018 [25,26]. This decrease is clearly related to the universal adolescent vaccination, with coverage fluctuating between 75% and 90% [20,21], and to the universal vaccination of infants, whose coverage in 2018 was 98.1% and in 2022 was 98.2% [22,27], higher than the 90% hepatitis B vaccination coverage in children stated by WHO Global Health Sector Strategy on viral hepatitis [28]. Our study points to an important impact of adolescent vaccination on the HBV incidence of the population as a whole, showing that the decrease was higher in regions that started adolescent vaccination earlier, in 1991–1993, rather than later, in 1994–1996.

The 2017 Spanish national seroprevalence survey data confirm a high HBV vaccination coverage, with HBV surface antibody (anti-HBs) prevalence rates of 85.2%, 45.9%, and 74.7% in the age groups of 2–5 years, 10–14 years, and 20–29 years, respectively [24]. Although results were not statistically significant, the lower prevalence of anti-HBc in the 35–39 age group (i.e., persons who had the opportunity to be vaccinated as adolescents) in regions that implemented vaccination earlier points to a positive cohort effect of the adolescent vaccination programme.

The benefits of universal adolescent vaccination have been reported elsewhere, in studies by Gidding et al. [29] in Australia, where adolescent vaccination was recommended in 1996 and the universal vaccine was introduced in 2000, and by Sun et al. [30] in China, where HBV immunization schemes were not established until 1990.

In a Catalan retrospective cohort study assessing HBV incidence in 2000–2014, the population prevented fraction in the cohorts who were vaccinated in adolescence was 64.56% (95% CI 60.45–68.66) [20]. In their study of all acute cases reported from 1992 to 2007 to investigate the relationship between acute HBV incidence reduction and universal preadolescent vaccination, Oviedo et al. [31] confirmed that HBV incidence declined overall after the universal adolescent vaccination programme was introduced but increased in male immigrants of working age, highlighting that in addition to the universal vaccination program the vaccination of risk groups, especially immigrant, should be strengthened currently.

In Catalonia, the first Spanish region to implement universal adolescent vaccination, the decision was based on analysing the possible impact of universal adolescent vaccination (in addition to the vaccination of high-risk groups and the newborns of HBsAg-positive mothers) and considering cost-effectiveness analyses for different strategies [32]. Other authors have also pointed to the immediate health benefits of universal adolescent vaccination, particularly in preventing new HBV cases [33].

Our findings confirm that the overall HBV incidence was lower in regions that implemented universal adolescent vaccination earlier, in 1991–1993, rather than later, in 1994–1996. Incidence differences in regions that started vaccination earlier rather than later were higher in the 20–29 and 30–39 age groups, suggesting that sexually transmitted HBV (the main transmission mechanism in our series) was better controlled in the former regions. In Spain, in the 20-year period of 1988–2008, 40 HBV outbreaks were reported involving 114 cases, and in at least 8 outbreaks, sexual activity was identified as the mechanism of transmission [25].

Laurence et al. [33] pointed out that unprotected anal intercourse is more frequent in adolescent and young adult homosexual and bisexual men, and so the benefits of vaccination before the initiation of sexual activity are especially evident. Few studies describe specific HBV outbreaks in men who have sex with men (MSM), unlike the case of hepatitis A, for which outbreaks typically affect MSM [34,35]. Those studies of HBV in MSM include Shankar et al. [36], who reported a small outbreak of 5 cases in eastern England in 2015, and MacDonald et al. [37], who reported 18 cases in Scotland in 2020–2022. Because the sexual transmission mechanism for HBV is important in low-prevalence countries, it is possible that some sexual transmission-related outbreaks are underdetected or underreported, a possibility supported by our finding of a higher incidence in men than in women.

The benefits of adolescent vaccination have also been observed in a study by Boccallini et al. [38] in Italy, where universal HBV vaccination was introduced in 1991 in a double-cohort strategy, for newborns and 12-year-olds; 30 years on, these authors reported (aside from the important health benefit of reduced incidence) considerable economic benefits in relation to costs associated with both acute and chronic HBV, with net savings of EUR 396,494,926 from a national health system perspective and EUR 482,577,670 from a societal perspective. Indeed, the scientific evidence available today suggests that HBV vaccination, irrespective of HBV endemicity, is one of the most cost-effective public health interventions available [39].

There is some uncertainty regarding long-term protection after infant vaccination that might warrant reconsideration of the infant priming regime to favour schedules with more doses of antigen, the use of adjuvants, or more delayed final doses, as with the current schedules with doses at 2, 3, 4, and 11–15 months, in addition to a birth dose for high risk infants, as recommended by the WHO [1]. Our data suggest that the benefit of universal vaccination is maintained over time.

Our results are similar to those reported by Romano et al. in a study carried out in Italy, where the overall incidence per 100,000 inhabitants of acute hepatitis B fell from 5 in 1990 to 1.6 in 2006. This decline was even more striking in people aged 15–24, in whom the morbidity rate per 100,000 dropped from 17 to under 0.5 in the same period, representing a 34-fold decrease. As in our study, the incidence rate of acute hepatitis B was under 2 per 100,000 inhabitants, was higher in males than in females and most infections occurred in people aged over 30, who had not been vaccinated. The authors concluded that their study provided evidence that since a strong immunological memory may persist after immunization, no booster doses of vaccine are required to maintain long-term protection in immunocompetent individuals [40].

There is no information of genotypes in reported cases of acute hepatitis B infection, but some Spanish studies report that genotype D was the most frequent, followed by genotype A during the periods 2001–2002 [41] or 2000–2016 [42]. However, despite the association between HBV genotypes and variable disease courses, liver disease severity, and treatment outcomes [1,43,44,45], several authors state that there is cross-genotype protection, and therefore the current HBV vaccines containing HBsAg generated from genotype A are effective in preventing infections caused by all known HBV genotypes [11,43,44,46].

Some authors have maintained that vaccination in infancy is still protective in adolescence. A study in Germany by Schwarz et al. [39] reported that, following the administration of a challenge dose of monovalent HBV vaccine (to stimulate exposure to natural infection), over 90% of adolescents showed protection against HBV; furthermore, a strong anamnestic (memory) response to the challenge dose was observed in most of the study participants, with no safety concerns identified. The results of a study by Hefele et al. [47] also confirm reduced HBV infection in adolescents vaccinated in infancy. This would suggest that immunization in the first year of life prevents HBV infection in adolescents and young adults.

While WHO does not currently recommend a booster dose for any age group, booster vaccination for adolescents might eventually prove useful, not only in reinforcing and extending protection in individuals immunized as infants but also in eliminating HBV. The fact that substantial differences exist in measures of residual protection among teenagers after infant or adolescent HBV vaccination warrants close ongoing scrutiny of whether important differences could emerge in long-term protection with or without booster vaccination [48]. A US study by Le et al. [49] reported a significant decrease in vaccine-induced immunity rates over time, which would point to the need for surveillance and for possible booster doses for individuals with undetectable anti-HBs levels on reaching adulthood, especially when at increased risk of infection through unprotected sex or injected drug use.

One adolescent booster dose can protect high-risk individuals against chronic HBV infection in adulthood. The individuals born to HBsAg-positive mothers and protected during childhood by HBV vaccination are still at high risk of HBV breakthrough infections. Adolescent booster vaccination might therefore be appropriate for high-risk individuals found to have no detectable serum anti-HBs [50]. Currently, however, the overall impact of vaccine-escaping mutants seems to be low and so does not pose a public health risk or suggest the need to modify established hepatitis B vaccination programmes [51]. No HBV breakthrough infection occur in the vaccinated individuals who had been protected in childhood by a neonatal series [50]. The study by Schwarz et al. in Germany reported lengthy antibody persistence against HBV in 14–15-year-old adolescents who received four doses of vaccine during infancy [39]. The study by Hefele et al. carried out in the Lao People’s Democratic Republic documented a sizable and lasting reduction in HBV infection in adolescents aged 11–18 years born after the introduction of an infant HBV vaccination programme [47]. The findings of a systematic review further support the importance of the WHO target of 90% hepatitis B vaccination coverage in infants [52].

An Italian study of anti-HBs seroprevalence by Zanella et al. [53] reported that the highest positivity rate was in children aged 1–5 years, with the absence of anti-HBc highlighting the vaccine-acquired immunity of the enrolled subjects. It was also confirmed that HBV circulation in children and adolescents was dramatically reduced 27 years after the implementation of a mandatory universal vaccination program.

In a Spanish seroprevalence study of blood-donor recipients of solid organ transplants and of at-risk persons (receiving dialysis, attending sexually transmitted infection centres, with hepatitis C or human immunodeficiency virus infection, and healthcare workers), Soriano et al. [54] observed that serum markers were absent in 37% of donors and 63% of at-risk individuals, indicating potential HBV susceptibility; the authors recommended that populations should undergo HBV serological testing at least once, irrespective of any HBV exposure risk, so that susceptible individuals can be vaccinated following the strategy that had been adopted in the United States until 2023. In the USA, HBV vaccination is routinely recommended for all people aged 19–59 years, independently of risk factors [55], and persons aged 60 or older with no known risk factors for hepatitis B infection may receive a hepatitis B vaccine series [56]. A study in the USA reported that universal HBsAg screening for adults was not only cost-effective but was probably cost-saving compared to other selective screening alternatives [57].

Our study has some limitations. First, the data on HBV exposure factors are incomplete because the systematic collection of these data only started in 2014 and so the available data only reflect some cases. Second, HBV vaccination coverage data for the Spanish regions are not available for the entire study period, although coverage data as reported by the Ministry of Health were included in our discussion of results. In contrast, one strength of this study is that more than 91% of acute hepatitis B cases [26] were tested for immunoglobulin M anti-HBc, meaning that we did not experience any difficulty in differentiating an episode of acute hepatitis B from an acute exacerbation of chronic HBV, as reported by some authors [58].

## 5. Conclusions

There are two main findings in this study. The first is that following the introduction of systematic vaccination according to the CISNS Recommended Immunization Schedule, acute HBV incidence decreased in all the Spanish regions. The second is that incidence in young adults decreased more in regions that began adolescent vaccination earlier (1991–1993) rather than later (1994–1996). We conclude that maintaining high universal vaccination coverage in the first year of life and in at-risk groups is necessary to achieve HBV elimination by 2030. Studies to confirm the duration of vaccine-induced immunity and to confirm that the current vaccines protect against the most prevalent genotypes are needed to ensure that adult populations remain adequately protected.

## Figures and Tables

**Figure 1 vaccines-12-00488-f001:**
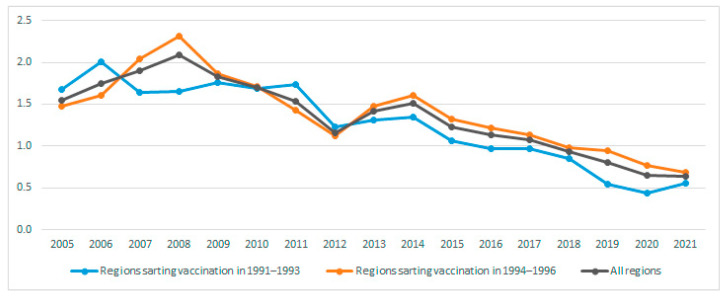
Acute hepatitis B incidence in 2005–2021 overall and for Spanish regions starting adolescent vaccination in 1991–1993 and regions starting adolescent vaccination in 1994–1996.

**Figure 2 vaccines-12-00488-f002:**
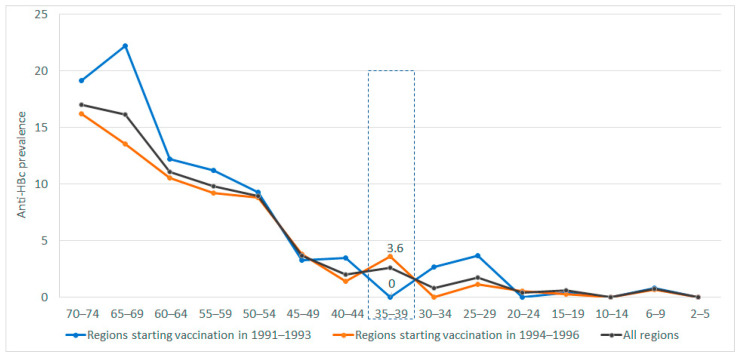
Anti-HBc prevalence in 2017–2018 in Spanish regions starting adolescent vaccination in 1991–1993 and regions starting adolescent vaccination in 1994–1996.

**Table 1 vaccines-12-00488-t001:** Acute hepatitis B incidence per 100,000 persons overall, by sex and age group in Spanish regions that started adolescent vaccination in 1991–1993 and in regions that started adolescent vaccination in 1994–1996.

	Overall				Men				Women			
	Adolescent Vaccination Started 1991–1993	Adolescent Vaccination Started 1994–1996	IRR (95% CI)	*p* Value	Adolescent Vaccination Started 1991–1993	Adolescent Vaccination Started 1994–1996	IRR (95% CI)	*p* Value	Adolescent Vaccination Started 1991–1993	Adolescent Vaccination Started 1994–1996	IRR (95% CI)	*p* Value
**All ages**												
**2005–2013**	1.628	1.670	0.97 (0.93–1.03)	0.32	2.484	2.243	**1.11 (1.04–1.18)**	**<0.001**	0.778	0.913	**0.85 (0.77–0.94)**	**0.002**
**2014–2021**	0.838	1.080	**0.76 (0.72–0.83)**	**<0.001**	1.250	1.636	**0.76 (0.70–0.83)**	**<0.001**	0.441	0.552	**0.80 (0.70–0.92)**	**0.001**
**20–29 years**												
**2005–2013**	2.097	2.421	**0.87 (0.77–0.98)**	**0.02**	2.859	2.838	1.01 (0.87–1.17)	0.92	1.287	1.636	**0.79 (0.63–0.98)**	**0.03**
**2014–2021**	0.716	1.003	**0.71 (0.56–0.90)**	**<0.001**	0.993	1.460	**0.68 (0.51–0.90)**	**0.006**	0.432	0.555	0.78 (0.50–1.21)	0.26
**30–39 years**												
**2005–2013**	3.001	2.884	1.04 (0.95–1.14)	0.38	4.618	4.055	**1.14 (1.03–1.26)**	**0.01**	1.201	1.282	0.94 (0.77–1.14)	0.52
**2014–2021**	1.285	1.741	**0.74 (0.63–0.86)**	**<0.01**	1.934	2.604	**0.74 (0.62–0.88)**	**<0.001**	0.637	0.873	**0.73 (0.54–0.99)**	**0.04**
**40–49 years**												
**2005–2013**	2.425	2.373	1.08 (0.97–1.21)	0.14	3.695	3.371	1.10 (0.97–1.24)	0.14	1.105	1.084	1.02 (0.81–1.27)	0.86
**2014–2021**	1.474	1.624	0.91 (0.79–1.04)	0.16	2.194	2.563	0.86 (0.73–1.00)	0.05	0.730	0.684	1.07 (0.81–1.41)	0.64

## Data Availability

The dataset is available from the corresponding author upon reasonable request.

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
