# Peer review of "Impact of the Universal Implementation of Adolescent Hepatitis B Vaccination in Spain"

_vaccines, 2024, doi:10.3390/vaccines12050488_

Round 1
Reviewer 1 Report
Comments and Suggestions for Authors
The aim of this study was to analyse the impact of universal adolescent HBV vaccination programme on incidence of acute hepatitis B in different regions of Spain. The universal adolescent (10-14 years) was introduced in 1991-93 in some regions and in 1994-96 in other regions of Spain.
The author reported an incidence decrease of acute HBV infection between 2005-2021 in all Spanish regions; however, the decrease of incidence was lower among young adult population in regions that began HBV vaccination in 1991-93.
Lines 40-41 and 72-73: The authors should report that currently the WHO recommends the hepatitis B vaccination as soon as possible after birth.
Line 111: as reported in Results, please add: “….the 20-29, 30-39 and 40-49 age groups…”
Figure 1 and line 137: name the axis of the figure and report “cases per 100.000 persons”
Table 1, line 147: Revise the caption by adding “cases per 100.000 persons” and “...incidence overall and by sex and group of age…”
Results: the authors must report whether the data on the coverage rate varied between the regions involved in the study.
Line 165: the authors should also report findings regarding the 40-49 age group.
Line 169: authors should report and comment on the higher prevalence of anti-HBc in the 23-30 age group in regions that began vaccinating in 1991-93.
Discussion: the discussion should be revised by focusing on the impact of adolescent hepatitis B vaccination and not reporting results of the impact of HBV neonatal vaccination.
Author Response
The aim of this study was to analyse the impact of universal adolescent HBV vaccination programme on incidence of acute hepatitis B in different regions of Spain. The universal adolescent (10-14 years) was introduced in 1991-93 in some regions and in 1994-96 in other regions of Spain.
The author reported an incidence decrease of acute HBV infection between 2005-2021 in all Spanish regions; however, the decrease of incidence was lower among young adult population in regions that began HBV vaccination in 1991-93.
Lines 40-41 and 72-73: The authors should report that currently the WHO recommends the hepatitis B vaccination as soon as possible after birth.
We agree. We have added this in the Introduction.
Line 111: as reported in Results, please add: “….the 20-29, 30-39 and 40-49 age groups…”
We have added the incidences rate for the 40-49 age group.
Figure 1 and line 137: name the axis of the figure and report “cases per 100.000 persons”
We have added “Cases per 100,000 persons” in the axis of the figure.
Table 1, line 147: Revise the caption by adding “cases per 100.000 persons” and “...incidence overall and by sex and group of age…”
We have changed the caption in Table 1.
Results: the authors must report whether the data on the coverage rate varied between the regions involved in the study.
Unfortunately, we do not have these data. This is stated in the limitations of the study, “HBV vaccination coverage data for the Spanish regions are not available for the entire study period, although coverage data as reported by the Ministry of Health were included in our discussion of results”.
Line 165: the authors should also report findings regarding the 40-49 age group.
We have added a paragraph with the results for the 40-49 age group.
Line 169: authors should report and comment on the higher prevalence of anti-HBc in the 23-30 age group in regions that began vaccinating in 1991-93.
We have added a paragraph with the results for the 25-34 age group. However, in 2017 aged everyone under 34 years was vaccinated, both people in the regions that started vaccination in 1991-93 and those that started later.
Discussion: the discussion should be revised by focusing on the impact of adolescent hepatitis B vaccination and not reporting results of the impact of HBV neonatal vaccination.
We have included information on adolescent vaccination coverage in the Discussion.
Reviewer 2 Report
Comments and Suggestions for Authors
This is a reasonably acceptable paper but there are some misleading statements and missing data and discussions that need to be addressed.
1) Lines 43-5 seem to imply that someone can get HBV from casual contact with bodily fluid. That is totally misleading. The bodily fluid has to enter the body.
https://www.hhs.gov/hepatitis/learn-about-viral-hepatitis/hepatitis-b-basics/index.html
2) There is no discussion of the various HBV serotypes. This important as certain vaccines may be more effective on certain serotypes
https://www.ncbi.nlm.nih.gov/pmc/articles/PMC4448583/
3) There is no mention of the types of vaccines used during the two periods. Were they more effective agains certain serotypes? If so which ones? Which vaccine is generally more effective?
4) There is no mention of the prevailing serotypes during the periods.
5) The authors concluded that the vaccine effectiveness was over time and this accounts for their statistical results. But (3) and (4) have to be addressed before they can conclude such.
Author Response
This is a reasonably acceptable paper but there are some misleading statements and missing data and discussions that need to be addressed.
1) Lines 43-5 seem to imply that someone can get HBV from casual contact with bodily fluid. That is totally misleading. The bodily fluid has to enter the body.
https://www.hhs.gov/hepatitis/learn-about-viral-hepatitis/hepatitis-b-basics/index.html
Thank you for your comment. We have changed the sentence to now reflect reference 1 (World Health Organization. Hepatitis B vaccines: WHO position paper-July 2017. Wkly Epidemiol Rec 2017, 27, 369-392) and reference 4 (Thio, C.L.; Hawkins, C. Hepatitis B virus. In Principles and Practice of Infectious Diseases, 9th ed.; Bennett, J.E., Dolin, R., Blaser, M.J., Eds.; Elsevier: Philadelphia, PA, USA, 2020; pp. 1940–1963).
2) There is no discussion of the various HBV serotypes. This important as certain vaccines may be more effective on certain serotypes
https://www.ncbi.nlm.nih.gov/pmc/articles/PMC4448583/
We have included information on the different HBV serotypes in the Introduction and Discussion.
3) There is no mention of the types of vaccines used during the two periods. Were they more effective against certain serotypes? If so which ones? Which vaccine is generally more effective?
The vaccines used in Spain during the study period were Engerix and Twinrix from GSK and HBVaxPro from Merck, recombinant vaccines generated from genotype A (Safadi R et al. Efficacy of birth dose vaccination in preventing mother-to-child transmission of hepatitis B: a randomized controlled trial comparing Engerix-B and Sci-B-Vac. Vaccines 2021, 9, 331; Ogawa M et al. Comparison of hepatitis B vaccine efficacy in Japanese students: a retrospective study. Environ Health Prev Med 2019, 24, 80; Bannister EG, et al. Molecular characterization of hepatitis B virus (HBV) in African children living in Australia identifies genotypes and variants associated with poor clinical outcome. J Gen Virol 2018, 99, 1103-1114).
In the Discussion we have added a sentence explaining that several authors state that there is cross-genotype protection and, therefore, the current HBV vaccines containing HBsAg generated from genotype A are effective at preventing infections caused by all known HBV genotypes.
4) There is no mention of the prevailing serotypes during the periods.
We do not have information on genotypes in reported cases of acute HBV infection, but some Spanish studies report genotype D to be most frequent followed by genotype A during the periods 2001-2002 (Ref. 41) and 2000-2016 (Ref. 42). We have added this in the Discussion.
5) The authors concluded that the vaccine effectiveness was over time and this accounts for their statistical results. But (3) and (4) have to be addressed before they can conclude such.
In the revised manuscript we have added “and to confirm that the current vaccines protect against the most prevalent genotypes” in the Conclusions.
Reviewer 3 Report
Comments and Suggestions for Authors
Manuscript: "Impact of the universal adolescent hepatitis B vaccination implementation in Spain”, by Domínguez et al.
Here, the authors carried out a survey of a cohort of citizens in Spain, and crunch the numbers, which show that HBV vaccine works. I understand, but isn’t that expected? So, what is new here? The novelty part, if there is any – needs to be put to the front, first in the Abstract and then again at the Introduction. Otherwise, the whole Introduction section, describing HBV vaccine application in different parts of the world, appears unnecessary and aimless.
However, the survey itself is impressively detailed, collecting all the available parameters of the study population: age, sex, microregion, and history of prior exposure over a 17-yr period. The results are meticulously organized. The graphs and the Supplementary table are presented in a simple and comprehensible format. These are all appreciated, although the stratifications seemed to show nothing remarkable. Overall, the findings in this paper is useful for the local health authorities for HBV control plan, but they are not fundamentally educational.
To strengthen the paper, I suggest that the authors should provide a short conclusion of EACH study, not just a final Conclusion paragraph for the whole paper. This will allow the readers to be convinced of the importance of each aspect of the survey, and why they were done.
Author Response
Manuscript: "Impact of the universal adolescent hepatitis B vaccination implementation in Spain”, by Domínguez et al.
Here, the authors carried out a survey of a cohort of citizens in Spain, and crunch the numbers, which show that HBV vaccine works. I understand, but isn’t that expected? So, what is new here? The novelty part, if there is any – needs to be put to the front, first in the Abstract and then again at the Introduction. Otherwise, the whole Introduction section, describing HBV vaccine application in different parts of the world, appears unnecessary and aimless.
However, the survey itself is impressively detailed, collecting all the available parameters of the study population: age, sex, microregion, and history of prior exposure over a 17-yr period. The results are meticulously organized. The graphs and the Supplementary table are presented in a simple and comprehensible format. These are all appreciated, although the stratifications seemed to show nothing remarkable. Overall, the findings in this paper is useful for the local health authorities for HBV control plan, but they are not fundamentally educational.
To strengthen the paper, I suggest that the authors should provide a short conclusion of EACH study, not just a final Conclusion paragraph for the whole paper. This will allow the readers to be convinced of the importance of each aspect of the survey, and why they were done.
Thank you for your comments. We agree that results of the study can be useful for regional and national public authorities, but they can be also educational for people interested in vaccination strategies and public health interventions because it is demonstrated that early introduction of universal vaccination of preadolescents translates into lower acute HBV infection incidence.
We have restructured the Conclusions.
Reviewer 4 Report
Comments and Suggestions for Authors
Title: Impact of the universal adolescent hepatitis B vaccination implementation in Spain.
Manuscript ID: 2952853
I recommended that manuscript could be accepted with MINOR MODIFICATIONS:
ABSTRACT
- Include p value in each comparison with or without statistical differences.
- A final sentence about impact of this study and probable recommendation could be useful.
INTRODUCTION
- Specify if previous authors compare different vaccination strategies in Spain.
MATERIAL AND METHOD
- A clear definition of regions should be included. If it is possible, territories or provinces that are included into the studied regions could be positive.
Author Response
Title: Impact of universal adolescent hepatitis B vaccination implementation in Spain.
Manuscript ID: 2952853
I recommended that manuscript could be accepted with MINOR MODIFICATIONS:
ABSTRACT
- Include p value in each comparison with or without statistical differences.
Thank you for your comments. We have included p values in the abstract.
- A final sentence about impact of this study and probable recommendation could be useful.
We have included a final sentence in the abstract.
INTRODUCTION
- Specify if previous authors compare different vaccination strategies in Spain.
Unfortunately, we have not found any studies that compare different vaccination strategies in Spain.
MATERIAL AND METHOD
- A clear definition of regions should be included. If it is possible, territories or provinces that are included into the studied regions could be positive.
We have stated when each Spanish region began to vaccinate in Methods.
Round 2
Reviewer 2 Report
Comments and Suggestions for Authors
improvements seen in this version.